# Instrumented Timed Up and Go Test (iTUG)—More Than Assessing Time to Predict Falls: A Systematic Review

**DOI:** 10.3390/s23073426

**Published:** 2023-03-24

**Authors:** Paulina Ortega-Bastidas, Britam Gómez, Pablo Aqueveque, Soledad Luarte-Martínez, Roberto Cano-de-la-Cuerda

**Affiliations:** 1Health Sciences PhD Programme, International Doctoral School, Universidad Rey Juan Carlos, 28922 Madrid, Spain; 2Kinesiology Department, Faculty of Medicine, Universidad de Concepción, Concepción, 151 Janequeo St., Concepcion 4030000, Chile; 3Biomedical Engineering, Faculty of Engineering, Universidad de Santiago de Chile, Libertador Bernardo O’Higgins Av., Santiago 9170022, Chile; 4Department of Electrical Engineering, Faculty of Engineering, Universidad de Concepción, 219 Edmundo Larenas St., Concepción 4030000, Chile; 5Physiotherapy, Occupational Therapy, Rehabilitation and Physical Medicine Department, Universidad Rey Juan Carlos, 28922 Madrid, Spain

**Keywords:** instrumented timed up and go, risk of falls, elderly

## Abstract

The Timed Up and Go (TUG) test is a widely used tool for assessing the risk of falls in older adults. However, to increase the test’s predictive value, the instrumented Timed Up and Go (iTUG) test has been developed, incorporating different technological approaches. This systematic review aims to explore the evidence of the technological proposal for the segmentation and analysis of iTUG in elderlies with or without pathologies. A search was conducted in five major databases, following PRISMA guidelines. The review included 40 studies that met the eligibility criteria. The most used technology was inertial sensors (75% of the studies), with healthy elderlies (35%) and elderlies with Parkinson’s disease (32.5%) being the most analyzed participants. In total, 97.5% of the studies applied automatic segmentation using rule-based algorithms. The iTUG test offers an economical and accessible alternative to increase the predictive value of TUG, identifying different variables, and can be used in clinical, community, and home settings.

## 1. Introduction

Falls are accidental events in which people lose control of their center of gravity, where the effort to regain balance is insufficient [1]. In total, 25% of the elderly population suffer at least one fall per year [2], increasing to two falls when the age is greater than 70 years [3]. For this reason, the risk of falls (RoF) is a public health issue [4,5], being considered one of the main causes of serious injuries in elderlies and the third cause of death due to unintentional injury [3,4,5], causing a sedentary life, loss of functional capacity and a decrease in the quality of life [6]. Thus, the clinical guidelines of “The American and British Geriatric Societies” recommend asking elderlies over 65 years if they have suffered two or more falls, if they have been injured during a fall or if they perceive any difficulty walking or maintaining balance [3]. Therefore, it is important to detect risk factors of falls and balance early, to implement effective and specific preventive clinical strategies [7].

The main causes of falls are multi-factorial, including extrinsic factors related to the environment, intrinsic factors related to the person, and behavioral factors related to the activity [8]. The most common intrinsic factors are muscle weakness, balance deficits, and gait instability [7]. For this, proposals to identify people with RoF, and measure balance and gait [8]. In clinical practice, different tests and observational scales are used to measure static and dynamic balance, as well as gait in healthy subjects or with motor impairments [5]. One of the most widely used tests is the Timed Up and Go (TUG), which measures dynamic balance and functional mobility [9,10,11].

The TUG test is a simple test that can be applied in several environments. It was developed in 1991 by Podsiadlo et al. [12] as a timed modification of the ”Get up and Go” test. It consists of a circuit in which the subject must get up from a chair, walk three meters, turn around and walk back to the chair to sit on it again (Figure 1). The controlled variable is the total test duration in seconds, which is then correlated with the RoF [8,12,13,14]. This test presents a high inter-rater and intra-rater reliability, with values greater than 95% in the prediction of RoF in elderlies, people with stroke [15,16] and Parkinson’s disease (PD) [17]. Other advantages of TUG are the simplicity and duration of its application. Additionally, it requires minimal equipment and allows subjects with functional disabilities to perform the evaluation. However, one of the limitations is that it cannot objectively determine the risk in subjects with greater difficulty. Barry et al. [8] mentioned that a limitation in the predictive value of the TUG test could be explained that it evaluates balance in a general way, which could be improved with the addition of technological tools for movement analysis [8,13].

Nowadays, there are proposals in the literature that allow the instrumentation of TUG through the use of sensors to capture and analyze movement. This variant of the test is called Instrumented Timed Up and Go (iTUG) [13,18,19]. This instrumentation makes it possible to identify the postural transitions performed during the test and to segment TUG into different sub-phases for extraction of specific measures for each of the identified sub-phases [20,21]. However, in the scientific literature, different technological and algorithmic proposals have been presented for iTUG segmentation that differs in the type of technology used, segmentation algorithms and extracted features for applications of intervention, characterization and RoF prediction [13,18,19,22,23,24].

Current systematic reviews on the TUG test study the psychometric properties of the test [25,26,27], incorporate information on RoF assessment instruments [5] or evaluate it as an evaluation instrument for a specific intervention [28]. However, to the best of our knowledge, there are no reviews where the applicability of iTUG in older adults is explored, describing the technological elements of the different scientific proposals reported in the literature and the segmentation and feature extraction strategies for its application.

Thus, we present a systematic review that explores the evidence of the technological proposals for the segmentation and analysis of iTUG in elderlies with or without associated pathologies, answering the following questions: What are the technological elements used, the methodological variations, and the main variables extracted in the application of iTUG in elderlies? What are the clinical applicability and predictive value of iTUG in elderlies?

## 2. Methods

The present review follows the guidelines of the PRISMA guide for systematic reviews [29]. The following databases were reviewed: National Library of Medicine, National Center for Biotechnology Information (NIH), Pubmed.gov; IEEE Xplore Digital Library, Scientific Electronic Library Online (SciELO), Elsevier, and Web of Science (WOS).

On 19 April 2022, a first search strategy was evaluated using the command:

(“itug” [All Fields] AND ((“accidental falls” [MeSH Terms] OR (“accidental” [All Fields] AND “falls” [All Fields]) OR “accidental falls” [All Fields] OR “falling” [All Fields] OR “falls” [All Fields] OR “fallings” [All Fields]) AND (“risk” [MeSH Terms] OR “risk” [All Fields]))) AND (y_10[Filter]).

From this strategy, only 37 articles were obtained from the Pubmed database and no results were obtained in the other databases. Thus, on 19 May 2022, the search command was modified as follows:

((instrumented AND (y_10[Filter])) AND (((timed up and go AND (y_10[Filter])) OR (Timed up & go AND (y_10[Filter]))) OR (TUG AND (y_10[Filter ])) AND (y_10[Filter]))) AND (elderly AND (y_10[Filter])) Filters: in the last 10 years.

This last command was used for the rest of the databases, obtaining the final search as (instrumented) AND ((timed up and go) OR (Timed up & go) OR (TUG)) AND ( elderly), including studies between the years 2012 and 2022. Three authors tested the last search command in each database to determine the effectiveness of the search or any difference.

On 31 October 2022, a final search was performed to update the database and identify new studies that could meet the eligibility criteria.

For the selection of the studies, a conceptual definition of iTUG was determined, as well as the context in which the proposals were analyzed. Regarding the types of studies included, no methodological limitations were applied to carry out the selection by level of evidence. The target population of the studies consisted of elderlies with a mean age equal to or greater than 65 years with or without associated pathologies. The description of the eligibility criteria can be observed in detail in Table 1 and Table 2.

To avoid bias in the selection and analysis of the studies, the initial registration and screening of the articles were carried out with COVIDENCE^®^ (Melbourne, Australia). Initial screening by title and abstract was performed by two authors using a blind methodology, and differences were discussed in conjunction with a third author to resolve discrepancies. During this selection, scientific studies were considered whose titles contained the keywords: instrumented, timed up and go, falls or risk of falling elderly or older adults and the conceptual definition of iTUG or that the description of the technology allowed to identify an iTUG. If the studies are considered potentially eligible, even if they did not meet the strategy described above, their extended reading was performed to corroborate whether or not they met the eligibility criteria in Table 1.

The extended review was also conducted at COVIDENCE^®^ (Melbourne, Australia) by two authors, whose disagreements were resolved by a third author.

Finally, the information extraction of the final selected articles was performed by two authors of the study, where the data were recorded and stored in a registration form made with Excel (Microsoft 365^®^, Redmond, WA, USA), which considered the following data:Authors;Year/Country;Study methodology;Institutions;Inclusion criteria;Exclusion criteria;Participants;Age of participants;Number of participants;Number of analyzed participants;Gender distribution;Technology/Sensors used;iTUG implementation;TUG implementation;Raw data;Index, parameters, and variables extracted;Segmentation Algorithm;Main outcomes;Main results.

Any discrepancy was resolved with the participation of a third reviewer.

## 3. Results

From the initial search in the different databases, 497 studies were obtained, and 74 were removed because they corresponded to duplicates, leaving a total of 423 studies. During the title and abstract screening stage, 344 articles were excluded, and 79 entered the extended review stage to determine their eligibility, eliminating 34 studies. Finally, 5 studies were excluded in the information extraction stage, and 40 were selected for analysis and discussion. The detail of the selection process can be seen in the PRISMA diagram from Figure 2.

The presentation of results are presented according to five sections based on the information extracted from the selected studies, which include characteristics of the participants and methodological design of the selected studies; types of technologies, procedures and instrumentation used in the TUG test; algorithmic procedures for segmentation and extraction of iTUG features; features extracted from iTUG; and main clinical results from the selected studies (see Table 3).

### 3.1. Characteristics of the Participants

From the selected studies, it can be observed that most participants were elderlies without health issues [24,33,34,39,42,43,48,50,58,59,61,63,65,67]. In only four studies, they were described as community elderlies [32,36,41,68], and in only one of them, they were described as residence elderlies [69]. Thirteen studies included participants with a diagnosis of PD [11,18,31,40,44,45,49,55,57,60,64,70] and only one of them with a diagnosis of stroke [13].

In addition, seven studies included participants with other pathologies, such as hip arthroplasty [23], possible “idiopathic normal pressure hydrocephalus (iNPH) [46], frailty index greater than 3.9 according to the Fried scale [52,53,54], neuropathy peripheral [62] and dementia [21]. Table 4 shows the studies according to the participant characteristics, diagnosis and the total number of participants.

According to the eligibility criteria, the age of the participants had to be greater than or equal to 65 years (see Figure 3), and the distribution by gender mostly showed a tendency to a greater number of women than men. Figure 4 illustrates these tendencies.

### 3.2. Methodological Design of the Selected Studies

The selected studies vary in the type of study design; although the level of evidence declared no methodological limitations, it can be seen in Table 5 that the predominant methodological designs were cross-sectional [21,32,44,52,53,54,55,63,65,69] and exploratory studies [11,18,41,45,46,48,50,61,62,64], with ten articles each, followed by cohort studies [33,36,67,70], with four articles. A smaller number of studies were experimental [22,24,31], with three articles, and descriptive [34,68], cross-sectional [43,59], prospective [23,49] and longitudinal [58,60], with two articles each. Finally, there were observed studies with case-control [42], clinical-randomized [39], and pilot [37] methodologies, with one article each. Finally, two articles did not mention the type of study [40,57].

### 3.3. Types of Technology, Procedure and Instrumentation Used in the Timed Up and Go Tests

The iTUG test has been defined as the use of inertial sensors to achieve the segmentation of the test and extract characteristics from the identified sub-phases [20,45]. However, in this review, different technological proposals were found. Table 6 shows the number of studies by type of technology used to implement the TUG.

Several proposals use inertial sensors for the instrumentation of the test [11,18,21,33,39,40,41,42,43,44,45,49,55,58,59,60,61,62,63,64,67,68,69,70]. Most of the studies use commercial sensors such as the G-Sensor, BTS G Walk^®^ (BTS Bioengineering, Lombardia, Italy) [39,40,41,42,43,69], whose inertial unit consists of a sensor with a tri-axial accelerometer and a tri-axial gyroscope, with a maximum sampling frequency of 1000 Hz. Other commercial sensors used include the Opal Sensor (APDM wearable technologies, Portland, OR, USA) [39,59], MTX XSens sensors (49A33G15, Xsens, Enschede, The Netherlands) [44,45,49], Dynaport sensor (McRoberts technologies, the Hague, The Netherlands) [18,33,55,63,67], which integrates a tri-axial accelerometer and gyroscope with a sampling frequency of 100 Hz, Shimmer inertial sensor (Shimmer technologies, Dublin, Ireland) [58], LEGSys and BalanSens (BioSensics, Boston, MA, USA) [70], mHT (mHealth Technologies, Bologna, Italy) [60,65], PAMSys inertial sensor (Biosensics, Newton, MA, USA) [62] and the tri-axial inertial sensor (Balance THETAmetrix, Portsmouth, UK) [21].

On the other hand, some studies mention the use of up to 17 measurement units [11,61], being able to acquire data from each body segment during iTUG. Other studies propose the use of six sensors, one in the sternum, one in L3, two in both hips and two in both thighs [59], or located in L5, two in the front part of the leg below the knee, two in the lateral part of the arm and one in the sternum [45]. However, most studies propose the use of a single sensor that is generally located between the lower back between L4 and S1, depending on the protocol used by the investigators [18,21,33,40,41,42,43,55,63,67,68,69].

Regarding the use of smartphone inertial sensors as technology for iTUG, four studies mention the use of the iPhone 4 smartphone (Apple Inc., Cupertino, CA, USA), located at the lower back [34] and on the sternum [52,53,54]. Two studies used Samsung Galaxy smartphones [46,48] located on the lower back. Another study used a Huawei P8 smartphone (Huawei, Shenzhen, China) positioned on the lower back [32].

On the other hand, three studies included the use of insoles, two of which considered an insole with 4 FSR (force-sensing resistors), which were positioned to measure the distribution of force in the foot. Two FSRs were positioned on the heel, one medial and one lateral, and the other two were located on the first and fifth metatarsals approximately, in conjunction with a 3D accelerometer attached to the foot [22,31]. A single study included the eSHOE insole system, which consists of a pair of orthopedic insoles that includes tri-axial accelerometers, tri-axial magnetometers, and a tri-axial gyroscope, as well as a pressure sensor on the greater toe, first and fifth metatarsal heads [37]. Two studies have incorporated the use of a sensorized chair called aTUG (ambiental Timed Up and Go), which considers an integrated chair with environmental sensors, four force sensors and a laser bar [24,50].

As mentioned above, the iTUG allows the segmentation of the TUG test into different sub-phases related to the activities that the participants must perform when executing the test. In the included literature, differences have been found regarding the number of sub-phases described in the segmentation. Some proposals have included the segmentation into three sub-phases, which consider the activities sit to stand, walk and stand to sit [31] or standing, forward walking and turning [39]. On the other hand, segmentation proposals have been found in four sub-phases that, in general, analyze the phases sit to stand, walk, 180° turn, and stand to sit [23,33,34,41,63,64,70]. However, segmentation proposals differ at the moment when a transition or transfer is initiated, for example, sub-phases have been described as sit-to-walk, walk, first turn [32,36] and turn to sit [32,33,36], or considering the last sub-phase directly from walking to sitting [55]. Likewise, five sub-phase segmentation proposals have been found, which describe sit-to-stand, walk-to-stand, turn, walk-to-sit and sit [24,52,53,54,68]. In the other five sub-phase proposals, differences have been found in the last phase of TUG, also describing the turn-to-sit phase [34,62,65]. On the other hand, the segmentation into six sub-phases of TUG considers sit-to-stand, forward gait, 180° turn, backward gait, turn, and stand-to-sit [11,18,40,43,46,50,61,69]. A single study proposes segmentation into the following phases: sit-to-stand, gait, turn, stand-to-sit, the full duration of the last turn to sit, the interval between the end of the last turn and the start of the stand-to-sit sub-phase [67].

Podsiadlo et al. [12] indicated that the standard procedure of TUG is that the user must stay sit in a chair without armrests, stand up from the chair, walk forward a distance of 3 m, turn around a mark or cone in the three meters, walk back and sit down on the chair. Most of the studies used the conventional procedure described previously [18,21,22,23,24,31,32,33,36,37,40,41,44,46,48,50,55,58,59,62,63,64,65,67,68,69,70]. However, some proposals used the extended versions of TUG with a distance of 5 m [11,61], 7 m [43,44,45,49] and 10 m [11,52,53,54,61], maintaining the same activities requested in the 3-meter TUG. In addition, proposals include traditional TUG plus dual tasks during its execution [39,42]. Lastly, only one study used a 2-meter TUG proposal [57].

### 3.4. Algorithmic Procedures for Segmentation and Extraction of iTUG Features

One objective of the instrumentation of clinical trials is to obtain augmented information that allows automating the evaluation, complementing the evaluation results with quantifiable information. In general, the iTUG allows to extract information from different sub-phases or postural transfers, through a process called segmentation.

In this review, different segmentation proposals have been found using rule based algorithms—that used the patterns of the acceleration and angular velocity signals—and, in some cases, machine learning tools for the identification of postural transitions. Table 7 presents a summary of the algorithms used in the included studies and their characteristics.

A large number of algorithms have been proposed for the analysis of iTUG, corresponding to 27 studies that use rule-based algorithms for the automatic segmentation of iTUG, 1 algorithm that is based on Machine Learning, 9 that use segmentation strategies developed by companies such as BTS G-Studio and APDM Mobility Lab and two studies that did not propose segmentation strategies. This could be explained byanalyzing pre-established movement patterns for the iTUG sub-phases, where rule-based algorithms perform well. All these algorithms implement the segmentation after the execution of the test and not in real-time.

Seven studies used the algorithm of Weiss et al. [33], being the most applied algorithm for the segmentation of iTUG using inertial sensors on the lower back. Five studies used the BTS G-Studio platform, which also uses a segmentation strategy with a single sensor located in the lower back. Four studies use the APDM Mobility Lab system. It is unclear from the studies which segmentation strategy is used, since inertial sensors can be located in up to six different body segments in combination to perform the segmentation process. Three studies used the Mellone et al. algorithm [35], being the preferred algorithm for iTUG segmentation when the technological base consists of inertial sensors of smartphones located on the upper thorax. Three studies used the algorithm of Walgaard et al. [56], which, in addition to using the information on the acceleration and angular velocity of the back, uses spatial orientation data (inclination and rotation) to determine postural transitions. Three studies used the Salarian et al. algorithm [47] to determine postural transitions from waist accelerometry and waist angular velocity.

Other studies that use information from inertial sensors to automatically identify postural transitions differ from previous proposals with respect to the sensor’s location—see Silva et al. for the thigh [48], Beyea et al. for the upper body [51], Najafi et al. for the chest [62] and Mariani et al. for the foot [64].

Regarding studies that use other technologies for the instrumentation of TUG, they use algorithms created by the same authors, such as Ayena et al. [31] for acceleration and foot pressure sensors, Tan et al. [57] for Kinect video, Frenken et al. [24] for environmental sensors (chair with pressure sensors and laser) and Holzreiter et al. [38] for coordinates of infrared markers for motion capture, the latter being the only one that uses Machine Learning strategies for the automatic identification of postural transitions.

Finally, it can be observed that nine segmentation strategies automatically identify seven postural transitions, which allowed twenty-eight studies to segment iTUG into the six main sub-phases (standing, go walking, first turn, return walking, pre-sitting turn and sitting). Three segmentation strategies only identified six postural transitions, which allowed five studies to segment iTUG into five sub-phases, combining the pre-sitting turn and sitting stage in a single sub-phase. Ayena et al. [31] segmented TUG into two postural transitions, allowing the identification of the sub-phases of standing, walking and sitting of the TUG test. Silva et al. [48] only used their segmentation strategy to identify the postural transitions of the first half of the test (before the return march), which, however, can be replicated for the return stage. Mariani et al. [64] only identified the turning sub-phases of the TUG.

### 3.5. Features Extracted from iTUG

Although the first stage of the iTUG analysis begins with the segmentation of the sub-phases or identification of the postural transitions, different methodologies can then extract characteristics, indices or parameters that characterize them quantitatively and objectively. Table 8 summarizes the characteristics used in the reviewed articles.

As indicated in Table 8, 23 studies use the duration parameters of each sub-phase of the segmented TUG, being the most used characteristic in studies for the characterization of elderly populations, intervention strategies and application of risk prediction tools. The gait sub-phase is the most analyzed, with 20 studies that extract and use gait cycle spatiotemporal parameters. Moreover, 19 studies used the total time of the test as a predictor of the risk of falls, 17 studies used statistical characteristics in the time domain of the angular velocity and 9 studies used statistical characteristics in the time domain of the acceleration, since the latter is more used to establish the patterns of postural transitions in segmentation algorithms and estimation of spatiotemporal gait parameters. Fourteen studies used statistical descriptors of mobility ranges or spatial orientation of body segments, where the thoracic, cervical, lumbar and pelvic segments were the most used. Two studies used statistical properties of acceleration from its Fourier transform, being the least used.

### 3.6. Main Clinical Results from the Selected Studies

In this section, the main results provided by the articles included in this review will be presented; the information will be organized by the type of participant included in the study, described in Table 4.

In elderlies without associated pathologies, it has been shown that iTUG, through the different technological proposals, allows extracting characteristics automatically, providing significant information on the temporal and velocity variables of each sub-phase, allowing the identification of groups with big or low RoF [34,43,48]. In addition, it allows to evaluate responses to the different treatments for improving balance, cognition and performance in dual tasks [39]. Furthermore, it can be combined with rating scales and allows to identification sub-clinical gait impairments [33,42]. On the other hand, it allows objectively and quickly identifying postural transitions during iTUG, analyzing sensory deficits and assessing the performance of the vestibular and somatosensory system [50,65].

iTUG and its segmentation allow characterizing the main sub-phases, identifying alterations in its execution in elderlies, such as the specific performance of people in the pre-sitting turning and sitting sub-phases, and relating it to other variables, such as cognitive and motor function [21,63,67]. Furthermore, iTUG combined machine learning algorithms, such as linear regression models [58,59] and other algorithms [24,61], identifying the decrease in the balance of highly functional elderlies, with a precision of 70% and a relationship of 80% for the identification of poor balance in tests performed one year later using one or more arrays of inertial sensors [58,59,61].

In community elderlies, it was shown that iTUG has adequate precision when compared with tests and community clinical assessment scales, suggesting that iTUG is a fast, economical tool, easy to administer in person or remotely [32], allowing it to be a reliable option for objective, unsupervised and unobtrusive measurement of balance in the clinical setting or at home [68]. In addition, it has been used to determine the functional decline associated with aging and determine specific differences in sub-phases related to gender [36], in the relationship between motor functioning and global cognitive function [41] and in residence contexts, where a study was able to correlate features of some sub-phases with the age of the participants [69].

In elderlies with PD, some studies have shown that an instrumented insole with FRS located on the heel and toes with a reduced number of sensors may be sufficient to estimate the risk index for falls during walking, being able to calculate the variation in gait and balance parameters [22,31]. Furthermore, it has been shown that it is possible to reduce the number of inertial sensors from 17 to 4 to achieve test segmentation [11]. On the other hand, in elderlies with stroke, the use of instrumented insoles in rehabilitation was evaluated, being capable of measuring the level of gait and TUG, providing details in the movement analysis [37].

The iTUG test can also be used in clinical practice to assess the effects of pharmacology and physical therapy in people with PD, such as the effects that L-Dopa may have on gait parameters and freezing on gait (FOG), as well as quantify and measure FOG [40,44].

On the other hand, studies indicated that iTUG could be correlated with scores from different balance and gait scales, demonstrating that an instrumented scale can reveal deficits in turnings of people with PD (severe and mild), as well as instrumentation in stages with or without medication making it possible to predict falls [45,49].

Another study determined the intra-rater, inter-rater, and test-retest reliability of iTUG in people with PD, proving excellent to good for the total duration and turning durations [18].

Regarding variables identified for people with PD through the instrumentation of TUG, studies explored gait speed in the clinic and home and the execution of iTUG at a fast speed, demonstrating that the related parameters during walking and turning showed strong correlations with the stage of the disease and that the application of the iTUG procedure at a fast speed allows the identification of movement deficits in mild to moderate stages, while the correlation in the parameters of the standing and sitting phases could determine the level of automation of the movements and the kinematic parameters of iTUG can have the potential to reflect functioning in movement execution [55,70].

On the other hand, iTUG measurements obtained from trunk angular velocity during the turning and standing phases adequately reflect dynamic balance in people with PD [60]. In the study carried out with the mTUG through the Kinectic system for Xbox One, it was possible to determine that the length of the first step can be significantly associated with the motor analysis scale implemented [57].

In addition, another study detected four phases of the test in groups of people with PD with and without medication through the use of sensors in the shoe during the execution of iTUG, where temporal variables proved to be the most relevant ones [64].

Finally, it has been found that iTUG is useful to analyze the gait patterns during the execution of the test applicable to healthy elderlies or with different gait disorders analyzing the correlations of the go and return sub-phases [23,46]. In elderlies with a frailty index greater than 3.95 according to the Fried scale, it was possible to observe that there are differences in acceleration signals and angular velocities of the trunk, allowing a more sensitive differentiation between frail and non-frail groups than only the TUG duration variable, which is traditionally used [52,53,54]. In elderlies with peripheral neuropathy and diabetes, the iTUG test allows them to identify and monitor postural transitions [62].

## 4. Discussion

The TUG test is a common tool for assessing mobility and fall risk in older adults that uses the time to identify from a global perspective the RoF; however, it is not able, in general, to incorporate the information of each sub-task performed in the evaluation, such as standing, walking, turning and sitting. The iTUG is a modified version of TUG that incorporates wearable sensors to capture additional objective data about gait, balance and other factors that may contribute to RoF.

This systematic review aimed to examine the currently existing evidence on the segmentation and analysis proposals of iTUG, the type of technologies used, the variables acquired and how these measures allow the specific detection of impairments in older adults with or without associated pathologies.

Previous systematic reviews related to the traditional TUG have been found that account for the different uses of the test alone. On the one hand, there are reviews in which TUG has been included as an assessment tool to identify changes in therapeutic interventions [28] and as an instrument to measure RoF [2,5], showing the applicability of the test in different clinical contexts. On the other hand, reviews were found that were related to different psychometric properties of the traditional TUG in different populations and with good reliability and validity values [25,26,27], showing that TUG has stable relative sensitivity when applied to older people in community settings [5], excellent intra-rater and inter-rater reliability and good construct validity, and is sufficiently sensitive to detect small changes in basic functional mobility after stroke [16] and adequate reliability and validity in people with PD [25].

However, it has been shown to have limited ability to predict falls in older people at high risk [8], as well as inconclusive results in its ability to predict falls after stroke [16]. It has been recommended that its application with other RoF measurement tools could increase its predictive value [5]. It has also been mentioned that its predictive value increases when instrumented (iTUG) [8,25]. In addition, we found a systematic review that addresses the instrumentation through the use of inertial sensors of different scales used to measure the risk of falls, including TUG, whose results account for the different characteristics extracted, the positioning of the sensors and the predictive value of each of them on the risk of falls [71]. However, to our knowledge, no similar systematic reviews have been found addressing the specific outcomes described in the present review using iTUG. These results are consistent with what was found in the present review, pointing out the high number of characteristics and parameters that can be obtained through the use of technologies. It is important to highlight that although TUG has shown different conclusions regarding its psychometric properties [4,16,25] both in older people and in people with stroke and PD, it is a tool that is still considered in clinical guidelines, and is commonly used, probably because of its easy implementation and because it requires little equipment [4,8].

The results of this review show the wide variety of devices used for TUG instrumentation, the type of procedure used, the characteristics of the population and the parameters that can be obtained from it. It demonstrates that all iTUG proposals, to a greater or lesser extent, allow the extraction of characteristics and variables of the subject’s performance during the procedure, increasing the test’s objectivity and providing additional values to the total time.

During the review, several technological proposals for TUG instrumentation were identified. However, the largest number of proposals include the use of inertial sensors [11,18,21,32,33,37,39,46,52,53,54,55,59,60,61,63,64,65,67,68,69,70]. Most proposals used a single sensor on the lower back, which has been shown to identify specific spatiotemporal characteristics, biomechanical elements of the pelvis and gait events during the execution of TUG, since it is close to the center of gravity [72,73,74]. This is important since the reduced use of sensors simplifies the implementation of iTUG in clinical settings.

On the other hand, it is important to highlight that extended versions of TUG were found at 5 [11,61], 7 [43,44,45,49], and 10 meters [11,52,53,54,61], delivering more information about gait characteristics together with the features extracted from the different transitions. It has been reported that iTUG using inertial sensors in its extended 7-meter version for people with stroke has shown excellent test-retest reliability [75]. However, it is important to consider new RoF labels for different populations, since most studies have analyzed the psychometric properties of the 3-meter version.

The different segmentation proposals allow researchers and clinicians to select TUG sub-phases according to their desired measurement objectives, which have provided a better analysis of TUG performance, increasing its predictive value. It is interesting to note that the segmentation of six sub-phases allows the identification of the last turn and the final stand phase before sitting. With this number of phases, it is possible to identify specific problems in elderlies with balance or sensory impairments, or the use of compensatory strategies [11,18,40,43,46,50,61,67,69].

The variables obtained by iTUG can be analyzed from a clinical point of view and how this information contributes to therapeutic decisions within the rehabilitation process. For example, in people with PD, it has been shown that they present alterations in the turning sub-phases, mainly associated with freezing [44]. The analysis of turning during iTUG segmentation allows healthy subjects to be differentiated from people with different pathologies so that the variables that can be analyzed during the turn not only provide information on the effects of aging [13]. This is of great clinical relevance, since iTUG could diagnose certain impairments or dysfunctions that lead to a loss of functional capacity and early implementation of rehabilitation strategies. Furthermore, people with advanced Parkinson’s disease have longer performance times in the non-instrumented TUG [76]. Still, the use of iTUG has made it possible to identify at different stages of the disease the variables in each of the sub-phases that are most altered, thus increasing its sensitivity in early stages [19]. Even the turning variables during iTUG execution are much more important in the case of pathologies that consider asymmetry during gait, as in people with stroke and PD [13].

The variables obtained with iTUG related to trunk movement could be clinically important for people with stroke since, in people with hemiparesis, it has been reported that those subjects who have experienced one or more falls present a greater sway of the center of mass in mediolateral and anteroposterior directions than those subjects who have not experienced falls. In addition, they present a longer duration in performing the sitting sub-phase when standing during the non-instrumented TUG. This time has been reported as an indicator of RoF [77].

The instrumentation of TUG not only allows observing more variables than just the total time of execution of the test, but also allows identifying the existing problems in each sub-task, thus optimizing the planning of rehabilitation plans and their follow-up, both in clinical and community contexts.

However, due to a large number of studies and their variability in designs, as well as the diverse implementation strategies of iTUG used and the varied segmentation or sub-stages identified, it is difficult to compare studies and extrapolate their findings to clinical scenarios. In future studies, it may be considered to standardize an implementation strategy based on the original protocol described by Podsiadlo for iTUG [12], as well as analyzing the highest amount of segmentation (see Table 7). This would allow for identifying more specific characteristics of the phases of sitting to standing, turning and standing to sitting, or an extended version of iTUG that allows for more gait characteristics to be extracted [43,44,45,52,53,54]. On the other hand, this review did not identify any studies on the predictive validity of iTUG, which should be considered, since one of the major problems presented by the traditional TUG is its low capacity to predict falls in some populations [8,16]. Studies comparing and correlating iTUG with other clinical scales with greater predictive value should also be conducted.

On the other hand, some methodological limitations of the articles included in this systematic review are the diversity of study designs. However, differences are found between analyzed populations, not only in terms of their features, but also in terms of sample sizes, which could interfere with the extrapolation or generalization of the results. The reviewed studies focus on the RoF analysis with iTUG in populations between 65 and 75 years old (see Figure 3). Future studies should include balanced samples of older adults above 75, recognizing that mobility and static and dynamic balance change with aging.

This systematic review has some limitations. In the first place, only five databases were searched (Pubmed, SciELO, IEEE, WOS and Elsevier), so we do not know if there is evidence in other databases. Furthermore, the search only considered articles in English and Spanish, and did not include other languages. Moreover, this review considers iTUG in subjects over 65 years of age; therefore, evidence of the application of this test in other age ranges or with other clinical tests that measure RoF was omitted. Furthermore, from a methodological point of view, the wide variety of study designs included did not allow a meta-analysis process to be performed; however, it was possible to incorporate the greatest number of technological proposals used for TUG instrumentation, which allows us to meet the aims of this review.

Finally, the results of this review show the possibility of increasing the predictive value of TUG through instrumentation by providing a more significant number of characteristics and parameters related to the subjects’ performance, allowing for objectively identifying alterations associated with static and dynamic balance disorders. Furthermore, this instrumentation allows for the increased applicability of the test not only as a measurement tool, but also as a possible diagnostic or predictive tool for RoF, as well as continuous monitoring of users during the rehabilitation process and their reintegration into community settings. The incorporation of technology or instrumentation could not only be applied to TUG, but could also be incorporated into other assessment scales or tests, which, along with the development of mobile applications and telemonitoring, could expand access for users residing in remote areas and therapeutic teams that do not have all the tools for implementing rehabilitation processes. This opens up the possibility of future lines of research in the clinical validation processes of technological proposals in different contexts and populations, the description of their psychometric properties and lines of technological development and data processing.

## 5. Conclusions

The iTUG test provides objective evaluations and guides treatment, making it a valuable tool for assessing the risk of falls in older adults. In this systematic review, the most used technology was inertial sensors, and healthy elderlies and elderlies with Parkinson’s disease were the most analyzed participants. The algorithm proposed by Weiss et al. was the most used for automatic segmentation. The iTUG test offers an economical and accessible alternative to increase the predictive value of the TUG test, identifying different variables, and can be used in clinical, community and home settings. The review’s findings highlight the potential benefits of incorporating technological approaches to increase the predictive value of TUG and improve RoF assessments.

## Figures and Tables

**Figure 1 sensors-23-03426-f001:**
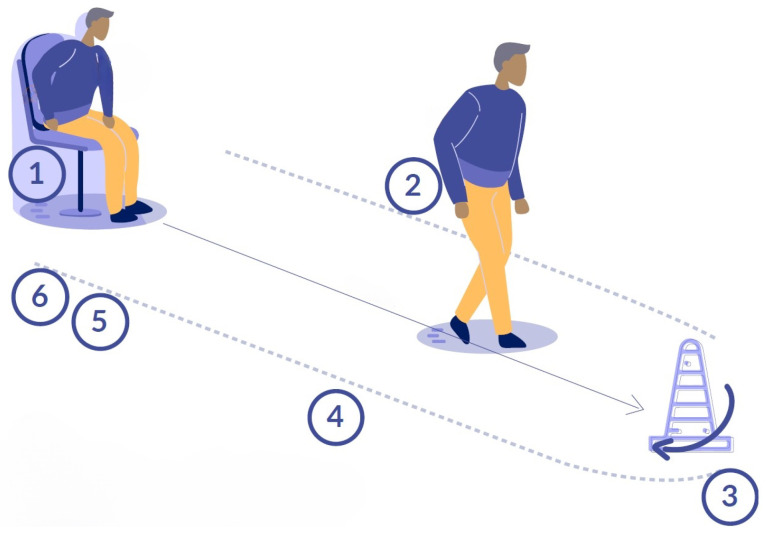
Timed Up and Go test with the different sub-phases after the most complete segmentation. (**1**) Standing. (**2**) Go Walking. (**3**) Three-meter turning. (**4**) Return Walking. (**5**) Pre-sitting turning. (**6**) Sitting.

**Figure 2 sensors-23-03426-f002:**
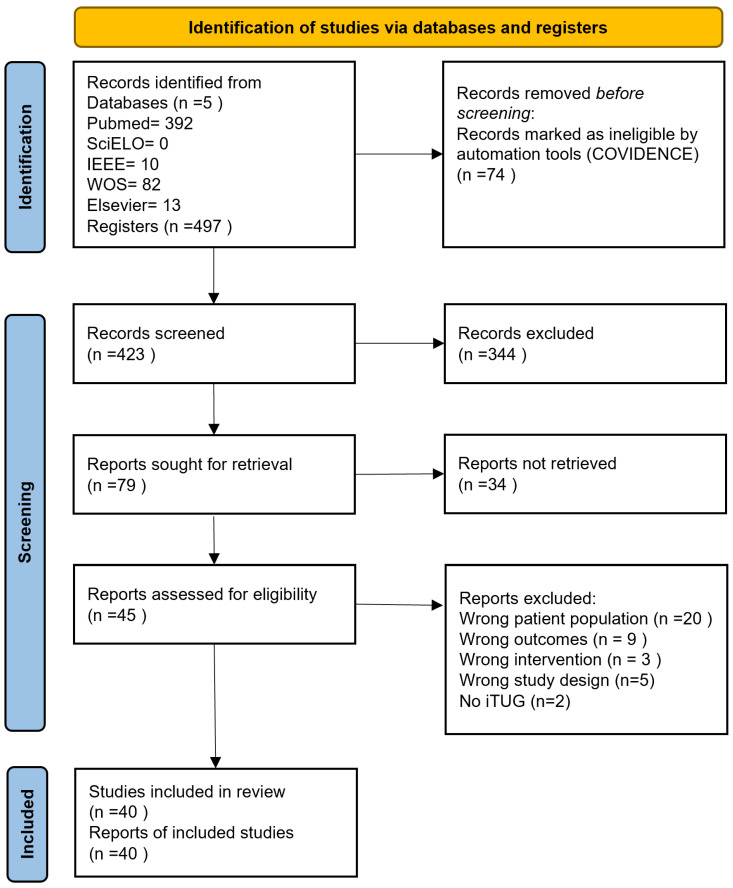
Systematic review screening process performed in this studio. Flowchart template extracted and modify from [30].

**Figure 3 sensors-23-03426-f003:**
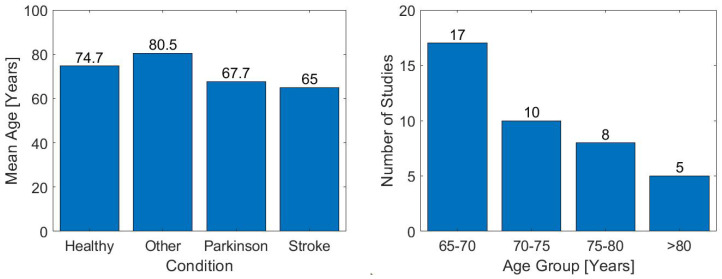
Distribution of the selected study participant’s ages, indicating the participant’s condition and the predominant general age range. In the participant’s condition plot, the “Other” category means elderlies with dementia, hip arthroplasty, diabetes and frail syndrome.

**Figure 4 sensors-23-03426-f004:**
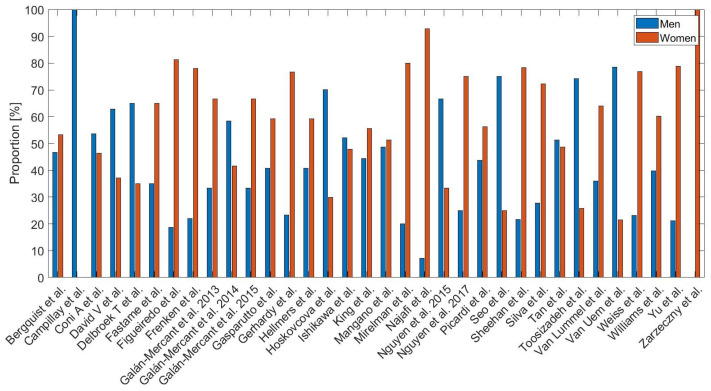
Proportion in percentages of the participants by sex from the studies selected.

**Table 1 sensors-23-03426-t001:** Eligibility criteria for study selection.

Criteria	Description
iTUG	Studies describe iTUG using technological support to enrich the test (cameras, inertial sensors, environmental sensors, pressure sensors, optoelectronic systems).
Index, parameters or variables	Proposals that, in addition to the total duration traditionally obtained in TUG, provide other variables or measurements that allow the identification of motor alterations in the participants.
Context	Proposals evaluated in community, clinical or academic settings.
Study methodology	Descriptive, experimental, quasi-experimental and proof-of-concept clinical studies were included that used validated commercial technology or new technologies whose applications were applied in the elderly population. No methodological limitations were applied to carry out the screening by the level of evidence.
Participants	Elderlies with a mean age equal or greater than 65 years, with or without associated pathology.
Language	Studies published in English or Spanish.
Study year	Studies published between 2012 and 2022.

TUG = Timed Up and Go; iTUG = Instrumented Timed Up and Go.

**Table 2 sensors-23-03426-t002:** Exclusion criteria for study selection.

Criteria	Description
Index, parameters or variables	Proposals that only provided the total value of TUG without demonstrating any other characteristic extracted, independent of declaring the use of any technology, were excluded.
Study methodology	Other narratives, bibliographic, systematic or scoping reviews were excluded, as well as “one-page” conference articles, abstracts, posters, letters to the editor and studies of iTUG psychometric properties validation.

TUG = Timed Up and Go; iTUG = Instrumented Timed Up and Go.

**Table 3 sensors-23-03426-t003:** Extensive description of iTUG methodology of all the studies included in the review.

Author	Methods	iTUG Implementation	Segmentation	Features	Main Results
Ayena et al., 2022 [22]	Experimental clinical study	Traditional 3-meter TUG procedure in people with PD. Segmentation in four phases: Standing, Go and Return Walking and Sitting. Insole with 4 FSR, one in the heel, the other in the medial and lateral foot, one in the first and the other in the fifth metatarsal. The tri-axial accelerometer is in the instep of the foot.	Automatic segmentation based on patterns of plantar pressures and accelerations. The research team developed the algorithm.	Gait index variability.	A reduced set of FSR allows the measurement of gait parameters for measuring the RoF in young elderly and PD.
Ayena et al., 2020 [31]	Experimental clinical study	Traditional 3-meter TUG procedure in people with PD. Segmentation in four phases: Standing, Go and Return Walking and Sitting. Insole with 4 FSR, one in the heel, the other in the medial and lateral foot, one in the first and the other in the fifth metatarsal. The tri-axial accelerometer is in the instep of the foot.	Automatic segmentation based on patterns of plantar pressures and accelerations. The research team developed the algorithm.	12 spatiotemporal gait parameters.	The risk of falls index calculated with a minimum number of sensors is a promising tool for real-time analysis, as it provides similar statistical information in risk analysis with a larger number of sensors.
Bergquist et al., 2020 [32]	Cross-sectional study	Traditional procedure of the 3-meter TUG procedure in communitary elderlies performing five repetitions. Segmentation into five phases: Standing, Go walking, 3-meter turning, Return walking and Sitting. Smartphone is located on the lower back.	Segmentation using acceleration and angular velocity signals with the algorithm by Weiss et al. [33]	78 features: Total iTUG duration. iTUG sub-phase duration. Spatial gait parameters in turning and walking sub-phases.	The iTUG model has a predictive ability similar to the scores of the CBMS and the battery of clinical tests with a significantly lower prediction error.
Campillay et al., 2017 [34]	Descriptive study.	Traditional procedure of the 3-meter TUG procedure in elderlies, selecting the best of three repetitions. Segmentation into five phases: Standing, Go walking, 3-meter turning, Return walking and Sitting. Smartphone is located on the lower back.	Segmentation using acceleration and angular velocity signals with the algorithm by Mellone et al. [35]	Total iTUG duration. iTUG sub-phase duration.	Duration of iTUG sub-phases measured through the smartphone IMU are highly reproducible.
Coni et al., 2015 [36]	Cohort study.	Traditional procedure of the 3-meter TUG procedure in communitary elderlies. Segmentation into five phases: Standing, Go walking, 3-meter turning, Return walking and Sitting. Smartphone located on the waist.	Segmentation using acceleration and angular velocity signals with the algorithm by Weiss et al. [33]	38 features: Total duration of iTUG; iTUG sub-phase duration. Statistical descriptors of angular velocity in the turning sub-phase: peak, mean, Root Mean Square Value (RMS). Spatial gait parameters: number of steps on walking and turning sub-phases.	A significant number of features can be derived from the sensor signals, which can be grouped into factors with clear clinical value, allowing the measurement of many mobility skills from one, and not just total time.
David et al., 2017 [37]	Pilot study.	Traditional procedure of the 3-meter TUG procedure in people with stroke. Segmentation into three phases: Standing, Walking and Sitting.	Segmentation using the spatial position of markers with Machine Learning (Neural Networks) [38]	Spatiotemporal gait parameters.	The system has proven to be capable of measuring the level of walking and TUG, providing details in the analysis of the movement and parameters compared with the current data, which is the total time measured manually.
Delbroek et al., 2017 [39]	Randomized clinical study.	Traditional procedure of the 3-meter TUG procedure in elderlies with visual tasks. IMU on ankles, wrists and chest. Segmentation into six phases: Standing, Go walking, 3-meter turning, Return walking, Pre-sitting turning, Sitting.	APDM Mobility Lab proprietary algorithm.	Total duration of iTUG. Spatiotemporal gait parameters. Statistical descriptors of angular velocity and accelerations.	All iTUG measurements improve with l-dopa, except sit-to-stand transition (duration and AP acceleration) and stand-to-sit transition (all parameters).
Dibilio et al., 2017 [40]	Does not specify.	Traditional procedure of the 3-meter TUG procedure in people with PD with visual tasks. The sensor is on the lower back at L4–L5. Segmentation into six phases: Standing, Go walking, 3-meter turning, Return walking, Pre-sitting turning, Sitting.	BTS G-Studio proprietary algorithm.	Freezing of gait. Total duration of iTUG. Duration per sub-phase of iTUG. Spatiotemporal gait parameters.	Low but significant correlations were found between motor scores (iTUG parameters) and global cognitive function.
Fastame et al., 2022 [41]	Exploratory study.	Traditional procedure of the 3-meter TUG procedure in communitary elderlies with visual tasks. The sensor is on the lower back at L5–S1. Segmentation into six phases: Standing, Go walking, 3-meter turning, Return walking, Pre-sitting turning, Sitting.	BTS G-Studio proprietary algorithm.	Duration and average speed of each sub-phase of iTUG.	It is a robust, stable and sensitive measurement tool for “movement smoothness” independent of the speed and duration of the movement, capable of identifying RoF independent of the day’s speed.
Figuiredo et al., 2020 [42]	case-control study.	Traditional procedure of the 3-meter TUG procedure in elderlies with visual tasks. The sensor is on the lower back at L5–S1. Segmentation into six phases: Standing, Go walking, 3-meter turning, Return walking, Pre-sitting turning, Sitting.	BTS G-Studio proprietary algorithm.	Total duration of iTUG. Duration per sub-phase of iTUG. Statistical descriptors of angular velocity by sub-phase. Statistical descriptors of degrees of orientation and inclination by sub-phase.	It is possible to identify significant differences in the temporal variables of the sub-phases and their speed, mainly in the elderly population.
Mangano et al., 2020 [43]	Transversal study.	Traditional procedure of the 3-meter TUG procedure and an extended 7-meter version in elderlies. The sensor is on the lower back at L1. Segmentation into six phases: Standing, Go walking, 3-meter turning, Return walking, Pre-sitting turning, Sitting.	BTS G-Studio proprietary algorithm.	Duration and average speed of each sub-phase of iTUG.	No differences were found between Parkinson’s patients with RoF and without RoF in stride length and speed parameters. Conversely, there was a difference in the time of double support. In addition, good correlations were found between the ABC Scale and iTUG scales.
Mancini et al., 2012 [44]	Cross-sectional study.	Extended 7-meter TUG in people with PD. The sensor is on the lower back and on the leg shank segment. Segmentation into six phases: Standing, Go walking, 3-meter turning, Return walking, Pre-sitting turning, Sitting.	APDM Mobility Lab proprietary algorithm.	52 features: Spatiotemporal gait parameters. Statistical descriptors of acceleration of the walking sub-phase. Acceleration frequency analysis of the gait sub-phase.	An instrumented scale can reveal deficits in the turns of patients with Parkinson’s (severe and mild), even though they have a normal score and walk correctly.
King et al., 2012 [45]	Exploratory study.	Extended 7-meter TUG in people with PD. The sensor is on the lower back, leg shank segment, arms and chest. Segmentation into six phases: Standing, Go walking, 3-meter turning, Return walking, Pre-sitting turning, Sitting.	APDM Mobility Lab proprietary algorithm.	Total duration of iTUG; Turning sub-phase duration. Spatial gait parameters: Step number during turning. Statistical descriptors of angular velocity during turning.	The system is useful for analyzing the gait patterns during TUG, applicable to healthy subjects or with different gait disorders by analyzing the correlations of the go and return signals.
Ishikawa et al., 2019 [46]	Exploratory study.	Traditional procedure of the 3-meter TUG procedure in elderlies with iNPH. Smartphone on the abdomen. Segmentation into six phases: Standing, Go walking, 3-meter turning, Return walking, Pre-sitting turning, Sitting.	Segmentation using acceleration and angular velocity signals with the algorithm by Salarian et al. [47].	Spatiotemporal gait parameters: Gait speed, cadence, stride time variability.	It could be correctly identified and with an error under the test transition elements in the predetermined order and in any other sequence of the movements or activities measured during TUG.
Silva et al., 2016 [48]	Exploratory study.	Traditional procedure of the 3-meter TUG procedure in elderlies. The sensor is in a pocket or attached to the thigh. Segmentation into three phases: Standing, Go walking, 3-meter turning.	Automatic segmentation algorithm using the integral of the angular velocity signal developed by the same research team.	Total duration of iTUG; Duration per sub-phase of iTUG; Spatial gait parameters: Step number during walking. Statistical descriptors of acceleration by sub-phase. Acceleration frequency analysis by sub-phase.	A strategy implemented in stages with and without medication in patients with Parkinson’s makes it possible to predict falls.
Hoskovcova et al., 2015 [49]	Prospective study.	Traditional 3-meter TUG and extended 7-meter TUG in people with PD. Use of five inertial sensors located in the leg, wrist and sternum areas.	Does not report segmentation strategy.	Spatiotemporal gait parameters: gait speed, cadence and stride time variability.	Using principal component analysis, it was observed that the characteristics that contributed the most to the discrimination with respect to the control group were in the phase of getting up, the chest flexion, the maximum obliquity of the chest and the maximum vertical velocity of the chest.
Hellmers et al., 2018 [50]	Exploratory study.	Traditional 3-meter TUG in elderlies. A belt with sensors in L4–L5 and an instrumented chair with four pressure sensors and a laser are used. Segmentation into six phases: Standing, Go walking, 3-meter turning, Return walking, Pre-sitting turning, Sitting.	Segmentation with the algorithm of Nguyen et al., 2015 [11].	Total duration of iTUG; Turning sub-phase duration. Statistical descriptors of acceleration by sub-phase.	There are some differences in the acceleration signals between groups of frail adults compared to non-frail adults by means of a smartphone placed on the chest.
Gasparutto et al., 2021 [23]	Prospective study.	Traditional procedure of the 3-meter TUG procedure in elderlies with hip arthroplasty. In total, 35 reflective markers were used according to the conventional gait model. Segmentation into five phases: Standing, Go walking, 3-meter turning, Return walking and Sitting.	Automatic segmentation with the algorithm by Bayea et al. [51].	33 features: joint mobility range descriptors by iTUG sub-phase: spine at thorax level and c7. The base of support by sub-phase. Spatial gait parameters: number of steps during walking and turning. Statistical descriptors of angular velocity by sub-phase of the thorax and pelvis.	This simple test could be appropriate for quantifying patient-specific deficits in function, and hence, guiding and monitoring post-operative rehabilitation in clinical settings.
Galán-Mercant et al., 2015 [52]	Cross-sectional study.	Extended 10-meter TUG in elderlies with the frail syndrome. Smartphone on thorax. Segmentation into five phases: Standing, Go walking, 3-meter turning, Return walking and Sitting.	Segmentation using acceleration and angular velocity signals with algorithms from Weiss et al. [33] and Salarian et al. [47].	Duration per sub-phase of iTUG. Statistical descriptors of acceleration by sub-phase. Statistical descriptors of angular velocity by sub-phase. Statistical descriptors of degrees of orientation and inclination by sub-phase.	An inertial sensor from the iPhone 4 can study and analyze the kinematics of the different sub-phases of the Extended TUG test in frail and non-frail elderly people.
Galán-Mercant et al., 2013 [53]	Cross-sectional study.	Extended 10-meter TUG in elderlies with frail syndrome. Smartphone on thorax. Segmentation into five phases: Standing, Go walking, 3-meter turning, Return walking and Sitting.	Segmentation using acceleration and angular velocity signals with algorithms from Weiss et al. [33] and Salarian et al. [47].	Statistical descriptors of acceleration by sub-phase. Statistical descriptors of angular velocity by sub-phase. Statistical descriptors of degrees of orientation and inclination by sub-phase. Total duration of iTUG. Duration per sub-phase of iTUG.	The inertial sensor found in the iPhone 4 is able to study and analyze the kinematics of the turning transitions in frail and physically active elderly persons. The accelerometry values for the frail elderly are lower than the physically active elderly, while variability in the readings for the frail elderly is also lower than the control group.
Galán-Mercant et al., 2014 [54]	Cross-sectional study.	Extended 10-meter TUG in elderlies with frail syndrome. Smartphone on thorax. Segmentation into five phases: Standing, Go walking, 3-meter turning, Return walking and Sitting.	Segmentation using acceleration and angular velocity signals with algorithms from Weiss et al. [33] and Salarian et al. [47].	Statistical descriptors of acceleration by sub-phase. Statistical descriptors of angular velocity by sub-phase. Total duration of iTUG. Duration per sub-phase of iTUG. Joint mobility ranges by sub-phase of iTUG (Thorax).	For the Extended TUG test, this device allows more sensitive differentiation between population groups than the traditionally used variable, namely time.
Van Uem et al., 2016 [55]	Cross-sectional study.	Traditional 3-meter TUG in people with PD. A belt with sensors in L4–L5. Segmentation into six phases: Standing, Go walking, 3-meter turning, Return walking, Pre-sitting turning, Sitting.	Segmentation using angular velocity signals and accelerations with the algorithm of Walgaard et al. [56].	Statistical descriptors of acceleration by sub-phase. Statistical descriptors of angular velocity by sub-phase. Total duration of iTUG. Duration per sub-phase of iTUG. Joint mobility ranges by sub-phase of iTUG (Lumbar).	Spontaneous physical activity at home and instrumented assessments in the clinic demonstrated strong discriminatory power in detecting impaired motor function in Parkinson’s disease.
Van Lummel et al., 2016 [18]	Exploratory study.	Traditional 3-meter TUG in people with PD. A belt with sensors in L4–L5. Segmentation into six phases: Standing, Go walking, 3-meter turning, Return walking, Pre-sitting turning, Sitting.	Segmentation using angular velocity signals and accelerations with the algorithm of Walgaard et al. [56].	Total duration of iTUG. Duration per sub-phase of iTUG.	Intra-rater, inter-rater and test-retest reliability of the individual components of iTUG was excellent to good for total duration and turning durations and good to poor for sitting durations and kinematics.
Toosizadeh et al., 2015 [53]	Cohort study.	Traditional 3-meter TUG in people with PD. Five units of measurement, one on each leg, one on each thigh, and one on the lower back. Segmentation into five phases: Standing, Go walking, 3-meter turning, Return walking, Sitting.	Segmentation using acceleration and angular velocity signals with the algorithm by Salarian et al. [47].	Statistical descriptors of acceleration by sub-phase. Statistical descriptors of angular velocity by sub-phase. Total duration of iTUG. Duration per sub-phase of iTUG. Joint mobility ranges by sub-phase of iTUG.	iTUG measurements obtained from trunk angular velocity during the turning and standing phases are adequate measures of dynamic balance in Parkinson’s disease. These measurements respond to rehabilitation, being able to detect improvement in patients after treatment.
Tan et al., 2018 [57]	Does not specify.	Reduced version at 2-meter TUG in people with PD. The Kinect camera is located 1.4 m from the test circuit.	Algorithm developed by the same research team.	Spatial gait parameters: Length of the first step. Total duration of iTUG. Turning sub-phase duration.	The only Kinect-derived variable significantly associated with the motor analysis scale was first step length.
Sheehan et al., 2014 [58]	Longitudinal study.	Traditional 3-meter TUG in elderlies. Sensors are located on each leg.	Does not specify a segmentation algorithm.	Spatiotemporal gait parameters. Spatial gait parameters. Statistical descriptors of angular velocity by sub-phase.	The quantitative values of iTUG and linear regression models allow us to identify decreases in the balance of highly functional older adults.
Seo et al., 2019 [59]	Transversal study.	Traditional procedure of the 3-meter TUG procedure in elderlies with visual tasks. IMU on ankles, wrists, lower back and chest. Segmentation into five phases: Standing, Go walking, 3-meter turning, Return walking, Sitting.	APDM Mobility Lab proprietary algorithm.	36 features: Spatiotemporal gait parameters. Statistical descriptors of angular velocity by sub-phase. Total duration of iTUG. Duration per sub-phase of iTUG. Joint mobility ranges by sub-phase of iTUG (thorax).	It was possible to predict the falls with 70.2% accuracy using the characteristics of the total duration of the test, standing characteristics, the maximum speed of the trunk in the sagittal plane and its range of movement in the horizontal plane during walking and the speed angular maxima during sitting.
Picardi et al., 2020 [60]	Longitudinal study.	Traditional 3-meter TUG in people with PD. The sensor on the lower back. Segmentation into five phases: Standing, Go walking, 3-meter turning, Return walking and Sitting.	Segmentation using the morphology of the acceleration signal with the algorithm of Mellone et al. [35].	Spatiotemporal gait parameters. Statistical descriptors of angular velocity by sub-phase. Total duration of iTUG. Duration per sub-phase of iTUG.	iTUG measurements obtained from trunk angular velocity during the turning and standing phases are adequate measures of dynamic balance in Parkinson’s disease, and these measurements respond to rehabilitation, being able to detect improvement in patients after treatment.
Nguyen et al., 2015 [61]	Exploratory study.	Extended 5-meter TUG and 10-meter TUG in elderlies. 17 IMUs in each body segment. Segmentation into six phases: Standing, Go walking, 3-meter turning, Return walking, Pre-sitting turning, Sitting.	Algorithm developed by the same research team.	36 features: Spatiotemporal gait parameters. Statistical descriptors of angular velocity by sub-phase. Total duration of iTUG. Duration per sub-phase of iTUG. Joint mobility ranges by iTUG sub-phase.	The proposed algorithm allows detecting and segmenting typical TUG activities using inertial sensors.
Nguyen et al., 2017 [11]	Exploratory study.	Extended 5-meter TUG and 10-meter TUG in eldelries. 17 IMUs in each body segment. Segmentation into six phases: Standing, Go walking, 3-meter turning, Return walking, Pre-sitting turning, Sitting.	Algorithm developed by the same research team.	Spatial gait parameters: number of steps during walking and turning. Statistical descriptors of acceleration by sub-phase. Statistical descriptors of angular velocity by sub-phase. Joint mobility ranges by sub-phase of iTUG.	The transferability of the segmentation methodology to Parkinson’s patients is demonstrated, using only 4 of the 17 initially predisposed sensors.
Najafi et al., 2013 [62]	Exploratory study.	Traditional procedure of the 3-meter TUG procedure in elderlies with peripheral neuropathy and diabetes with visual tasks. Sensor on thorax. Segmentation into five phases: Standing, Go walking, 3-meter turning, Return walking and Sitting.	Evaluation of postural transitions through accelerometer analysis. The algorithm was developed by the same research team.	Duration per sub-phase of iTUG.	The proposed system can identify and monitor postural transitions, accurately identifying subjects with a high RoF with potential use in monitoring older adults with diabetes.
Mirelman et al., 2014 [63]	Cross-sectional study.	Traditional 3-meter TUG in elderlies. Sensor on the lower back. Segmentation into five phases: Standing, Go walking, 3-meter turning, Return walking, Sitting.	Segmentation using the morphology of the acceleration and angular velocity signal with the algorithm of Weiss et al. [33].	Duration per sub-phase of iTUG.	Using a single sensor on the back during TUG can quantify mobility, facilitating the understanding of problems related to cognitive decline.
Mariani et al., 2013 [64]	Exploratory study.	Traditional 3-meter TUG in people with PD. IMU on the instep of the foot. Segmentation into three phases: Go walking, 3-meter turning, Return walking.	Segmentation using angular velocity signals and accelerations. The algorithm was developed by the same research team.	Duration per sub-phase of iTUG. Spatiotemporal gait parameters: step number and gait velocity.	All the sub-phases of the test could be detected in the control groups, with medication and without medication. Variations of the measured gait characteristics are observed, where the temporal variables are the most important within the TUG test.
Frenken et al., 2013 [24]	Experimental study.	Traditional 3-meter TUG in elderlies. No sensors on the participants. aTUG (ambient TUG) chair integrated with environmental sensors, four force sensors and an optical laser. Segmentation into fivex phases: Standing, Go walking, 3-meter turning, Return walking, Sitting.	Algorithm developed by the same research team.	Statistical descriptors of degrees of orientation and inclination by sub-phase. Duration per sub-phase of iTUG.	The proposed method is able to accurately compute the duration of the TUG components using only the force sensors and the laser range scanner.
Gerhardy et al., 2019 [65]	Cross-sectional study.	Traditional procedure of the 3-meter TUG procedure in elderlies. The sensor is on lower back. Segmentation into five phases: Standing, Go walking, 3-meter turning, Return walking, Sitting.	Segmentation using the morphology of the acceleration signal with the algorithm of Mellone et al. [35].	Total duration of iTUG. Duration per sub-phase of iTUG.	Total TUG time was strongly associated with vestibular and somatosensory system performance.
Weiss et al., 2013 [33]	Cohort study.	Traditional 3-meter TUG in elderlies. Sensor on the lower back. Segmentation into five phases: Standing, Go walking, 3-meter turning, Return walking, Sitting.	Segmentation using acceleration and angular velocity signals with algorithms developed by the same team [66].	Spatio temporal gait parameters: Stride speed, stride length, swing width, stride length. Statistical descriptors of degrees of orientation and inclination by sub-phase of rotation. Duration per sub-phase of iTUG.	People with mobility impairments have problems in the five sub-phases of TUG.
Weiss et al., 2016 [67]	Cohort study.	Traditional 3-meter TUG in elderlies with dementia. The sensor on the lower back. Segmentation into six phases: Standing, Go walking, 3-meter turning, Return walking, Pre-sitting turning, Sitting.	Segmentation using acceleration and angular velocity signals with algorithms developed by the same team [66].	Total duration of iTUG. Duration per sub-phase of iTUG.	Longer separations between the movements of the sub-phases and a longer overlap between turning and the stand-to-sit sub-phase are related to poorer cognitive and motor function and greater disability.
Williams et al., 2021 [21]	Cross-sectional study.	Traditional 3-meter TUG in elderlies with dementia. The sensor is on the lower back. Do not specify level of segmentation.	Segmentation using angular velocity signals and accelerations with the algorithm of Walgaard et al. [56].	Statistical descriptors of angular velocity by sub-phase.	The sensor and its algorithms were able to quantify the sub-phases of iTUG and demonstrate that age moderates differences in the performance of iTUG and its informal caregivers.
Yu et al., 2021 [68]	Descriptive study.	Traditional procedure of the 3-meter TUG procedure in communitary elderlies. Sensor on the lower back. Segmentation into five phases: Standing, Go walking, 3-meter turning, Return walking, Sitting.	Segmentation using acceleration and angular velocity signals with algorithms from Weiss et al. [33].	Statistical descriptors of acceleration by sub-phase. Statistical descriptors of angular velocity by sub-phase. Statistical descriptors of degrees of orientation and inclination by sub-phase.	The proposal is a reliable option for objective, unsupervised and unobtrusive balance measurement in a clinical or home setting.
Zarzeczny et al., 2017 [69]	Cross-sectional study.	Traditional procedure of the 3-meter TUG procedure in elderlies with visual tasks. Sensor on the lower back at L4–L5. Segmentation into six phases: Standing, Go walking, 3-meter turning, Return walking, Pre-sitting turning, Sitting.	BTS G-Studio proprietary algorithm.	Statistical descriptors of acceleration by sub-phase. Statistical descriptors of angular velocity by sub-phase. Statistical descriptors of degrees of orientation and inclination by sub-phase.	The performance of the functional tests is more dependent on the range of force developed than the maximum isometric force of the muscles of the lower extremities during acceleration when sitting is measured.

PD = Parkinson’s Disease; TUG = Timed Up and Go; FSR = Force Resistive Sensor; iTUG = Instrumented Timed Up and Go; CBMS = Community Balance and Mobility Scale; IMU = Inertial Measurement Unit; PD = Parkinson’s Disease; RMS = Root Mean Square; AP = Antero-posterior; RoF = Risk of Falls; ABC Scale = Activities-specific Balance Confidence Scale; iNPH = Idiopathic normal pressure hydrocephalus.

**Table 4 sensors-23-03426-t004:** Participants of the selected studies.

Subjects	Studies	Participants
Elderlies [24,33,34,39,42,43,48,50,58,59,61,63,65,67]	14	2448
Community elderlies [32,36,41,68]	4	521
Residence elderlies [69]	1	26
PD [11,18,31,40,44,45,49,55,57,60,64,70]	13	426
People with stroke [13]	1	35
Elderlies with other pathology [21,23,46,52,53,54,62]	7	376
Total	40	3822

PD = Parkinson’s disease.

**Table 5 sensors-23-03426-t005:** Type of studies reviewed.

Type of Study	Studies
Cross-sectional [21,32,44,52,53,54,55,63,65,69]	10
Exploratory [11,18,41,45,46,48,50,61,62,64]	10
Cohort [33,36,67,70]	4
Clinical-Experimental [22,24,31]	3
Descriptive [34,68]	2
Transversal [43,59]	2
Prospective [23,49]	2
Longitudinal [58,60]	2
case-control [42]	1
Clinical-Randomized [39]	1
Pilot [37]	1
Not mentioned [40,57]	2
Total	40

**Table 6 sensors-23-03426-t006:** Technologies and sensors used for iTUG.

Technology	Studies
Insoles [22,31,37]	3
Smartphone [32,34,36,46,48,52,53,54,65]	9
Inertial Sensors [11,18,21,33,39,40,41,42,43,44,45,49,55,58,59,60,61,62,63,64,67,68,69,70]	24
Opto–electronic System [23,55]	2
Xbox Kinect [57]	1
Instrumented Chair [24,50]	2

**Table 7 sensors-23-03426-t007:** Algorithms for iTUG segmentation identified from the selected studies.

Author	Proposal	Inputs	Transitions	Studies
Ayena et al. [31]	Rule-based.	Tri-axial foot accelerations. Plantar pressures.	S2W, W2S.	2
Weiss et al. [33]	Rule-based.	Tri-axial lower back accelerations and tri-axial lower back angular velocities	St, S2W, W2T, T2W, W2T, T2S, Si.	7
Mellone et al. [35]	Rule-based.	Tri-axial chest accelerations.	St, S2W, W2T, T2W, W2T, Si.	3
Holzreiter et al. [38]	Machine Learning.	On-body reflective infrared marker’s coordinates.	St, S2W, W2T, T2W, W2T, Si.	1
APDM Mobility Lab.	Propietary.	Tri-axial accelerations and tri-axial angular velocities from both wrists, shanks, lower back and/or chest.	St, S2W, W2T, T2W, W2T, T2S, Si.	4
BTS G-Studio.	Propietary.	Tri-axial lower back accelerations and tri-axial lower back angular velocities.	St, S2W, W2T, T2W, W2T, T2S, Si.	5
Silva et al. [48]	Rule-based.	Tri-axial thigh angular velocities.	St, S2W, W2T.	1
Salarian et al. [19]	Rule-based.	Tri-axial waist accelerations and tri-axial waist angular velocities.	St, S2W, W2T, T2W, W2T, T2S, Si.	3
Nguyen et al. [11]	Rule-based.	Tri-axial lower back accelerations and tri-axial lower back angular velocities.	St, S2W, W2T, T2W, W2T, T2S, Si.	3
Bayea et al. [51]	Rule-based.	Tri-axial upper body accelerations. Tri-axial upper body angular velocities.	St, S2W, W2T, T2W, W2T, T2S, Si.	1
Walgaard et al. [56]	Rule-based.	Tri-axial lower back accelerations. Tri-axial lower back angular velocities. Displacements and angles of lower back and feet.	St, S2W, W2T, T2W, W2T, T2S, Si.	3
Tan et al. [57]	Rule-based.	Kinect video.	St, S2W, W2T, T2W, W2T, T2S, Si.	1
Najafi et al. [62]	Rule-based.	Tri-axial chest accelerations. Tri-axial chest angular velocities.	St, S2W, W2T, T2W, W2T, T2S, Si.	1
Mariani et al. [64]	Rule-based.	Tri-axial foot accelerations. Tri-axial foot angular velocities.	W2T, T2W.	1
Frenken et al. [24]	Rule-based.	Pressure sensors on chair. Light laser distance.	St, S2W, W2T, T2W, W2T, Si.	1

St = Standing, S2W = Sit-to-Walk, W2T = Walk-to-Turn, T2W = Turn-to-Walk, W2T = Walk-to-Turn, T2S = Turn-to-Sit, and Si = Sitting.

**Table 8 sensors-23-03426-t008:** Main features extracted from the selected studies.

Feature	Relevant Parameters	Studies
Total iTUG duration.	Clinical Score.	19
Duration per sub-phase of iTUG.	Standing, walking (go and return), turning (3-meter and pre-sitting), sitting.	23
Spatiotemporal gait parameters.	Cadence, stride time, step time, stride length, step length, gait velocity, number of steps, double support time, stride time variability, gait index variability.	20
Statistical descriptors in the acceleration time domain by sub-phase.	Maximum value, minimum value, mean, standard deviation, kurtosis, root mean square value, entropy.	9
Statistical descriptors in the time domain of angular velocity by sub-phase.	Maximum value, minimum value, mean, standard deviation, kurtosis, root mean square value, entropy.	17
Statistical descriptors in time domain of mobility ranges in degrees per sub-phase.	Main joints measured: thorax (chest or sternum), cervical (C7), lumbar (L4–L5), pelvis.	14
Acceleration frequency domain descriptors per sub-phase.	Fast Fourier transform.	2

iTUG = Instrumented Timed Up and Go.

## Data Availability

Not applicable.

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
