# Peer review of "Instrumented Timed Up and Go Test (iTUG)—More Than Assessing Time to Predict Falls: A Systematic Review"

_sensors, 2023, doi:10.3390/s23073426_

Round 1

Reviewer 1 Report

This paper did a Systematic Review on the utilization of iTUG in predicting falls. The paper is organized according to the guideline of Systematic Review, and the obtained papers are presented. The paper is well written, and the results are helpful. Some minor revisions are needed to further improve the paper.

1.In the discussion, it’s better to introduce the main difference between utilizing iTUG in predicting falls and common TUG, eg. the concerns, techniques or targets etc.

2.It is not necessary to repeatedly show the complete spelling of abbreviations all over the paper.

3. Maybe, it will more helpful to make a summary on the imperfections provided by the papers, and give a simple prediction on the solutions.

4. The abstract and conclusion section has too many repeated items. It is better to rewrite the abstract.

Author Response

Dear Reviewer,

We appreciate the comments and suggestions sent. We detail below the modifications that have been made,

1. We introduce in the discussion the main difference between iTUG in predicting falls and common TUG, between lines 379-384.

2. In order to facilitate the reading of each table due to their length, abbreviations have been included in each table footnote.

3. We added some main limitations and possible solutions of the included articles, between lines 469-488. As well as the applicability of the results found in the present review, between lines 499-513.

4. Abstract and conclusion sections were rewritten, between lines 1-11 and 515-523, respectively.

Again, we appreciate the suggestions and hope that our modifications can improve the quality and understanding of our manuscript, in order to comply with the standard of the journal.

Sincerely,

Pablo Aqueveque Navarro, PhD.

Full Professor

Electrical Engineering Department,

Faculty of Engineering

Universidad de Concepción.

Concepción, Chile

Reviewer 2 Report

This paper systematically reviewed the studies regarding the evidence of the technological proposal for the segmentation and analysis of the iTUG in elderlies with or without pathologies. The review paper is valuable and contributive to the specific filed to guide the study direction for the related future investigations. As shown in this manuscript, the solid results are proposed and statistical findings are also presented for the state of the art of the iTUG use. I am pleased to say that this paper can be accepted for publication after minor corrections and revisions, as follows.

L11, In Abstract, in general, we do not include references or specific authors unless there are special cases.

L70, the term “Materials and Methods” may be not so appropriate for a review paper.

There are around 13 review papers in the references that were cited by this paper. Even though these references may not relate to the iTUG directly, it is suggested that authors should discuss about the results of these studies either in the Introduction or Discussion section, except in Table 5.

Figure 3. the bar indexed by “other” is a little strange compared to the other specific pathologies. The presentation is not clear. Please show the summary samples for each bar case. Moreover, if possible, please indicate the pathologies in the text for the “other” category.

In horizontal axis of Figure 4, Weiss et al. 2013, the year 2013 should be removed.

Author Response

Dear Reviewer,

We appreciate the comments and suggestions sent. We detail below the modifications that have been made,

1. Abstract Section were rewritten, between lines 1-11.

2. The term “Materials and Methods” were change to “Methods”, line 68.

3. We added extended in the discussion some results of the cited reviews, between lines 389-416. As well as some changes between lines 379-384, 469-488 and 499-513.

4. In Figure 3. We added at the footnote the specification of “other”, which means elderlies with dementia, hip arthroplasty, diabetes and frail syndrome, in line 160.

5. In horizontal axis of Figure 4, Weiss et al. 2013, the year 2013 has been removed, before line 161.

Again, we appreciate the suggestions and hope that our modifications can improve the quality and understanding of our manuscript, in order to comply with the standard of the journal.

Sincerely,

Pablo Aqueveque Navarro, PhD.

Full Professor

Electrical Engineering Department,

Faculty of Engineering

Universidad de Concepción.

Concepción, Chile

Reviewer 3 Report

This article presents a systematic literature review of a widely applicable fall risk assessment method in the elderly, namely the (Timed Up and Go – TUG), but in a variant incorporating additional technological devices (instrumented TUG - iTUG) allowing multiparametric analysis and refinement of medical assessments. Applying the PRISMA guidelines for systematic review, the authors formulated eligibility criteria for selecting valid papers by synthesizing logical search terms involving keywords and combinations such as: itug; timed up and go; tug accidental falls; fallings; risk; elderly. The search is focused on studies published between 2012 and 2022 in the following databases: Pubmed, IEEE Xplore Digital Library, Scientific Electronic Library Online (SciELO), Elsevier, Web of Science (WOS). The total number of articles identified in the initial search is 497. After applying additional filters to refine the selection, remove duplicates, and in-depth content analyses, the authors reduced the selection to 40 publications, according to the stated aspects of the study. The applied methodology is correct and covers a significant part of the publications with contributions in the field of study. The selection presents the state-of-the-art regarding to the applications of iTUG in clinical, community, and home settings. The authors present statistics and analysis regarding characteristics of the participants, methodological design, types of technology, procedure, and instrumentation used in the iTUG, algorithmic procedures for segmentation and extraction of iTUG features, main clinical results.

The abstract correctly present the content of the article. Reference sources are relevant to the content and are cited at appropriate places in the text.

A major shortcoming of this review is the lack of in-depth comparative analysis of the iTUG methodology with other approaches to fall risk assessment. The authors have limited themselves only to conclusions based on the specific applications described in the analyzed publications. In this regard, there is also a lack of vision for future development and application possibilities, which would be of interest to researchers, developers and users.

Author Response

Dear Reviewer,

We appreciate the comments and suggestions sent. In order to incorporate your appreciations and to highlight the importance of this topic we added at the end of the discussion section the applicability of the results obtain in this systematic review as well as future research lines, between lines 499-513, attached below.

“However, due to a large number of studies and their variability in designs, as well as the diverse implementation strategies of iTUG used and the varied segmentation or sub-stages identified, it is difficult to compare studies and extrapolate their findings to clinical scenarios. In future studies, it may be considered to standardize an implementation strategy based on the original protocol described by Podsiadlo for iTUG [12], as well as analyzing the greatest number of sub-segments of segmentation (see Table 7). This would allow for identifying more specific characteristics of the phases of sitting to standing, turning, and bipedal to sitting, or an extended version of iTUG that allows for more gait characteristics to be extracted. [43–45,52–54] On the other hand, this review did not identify any studies on the predictive validity of iTUG, which should be considered since one of the major problems presented by the traditional TUG is its low capacity to predict falls in some populations [8,16]. Studies comparing and correlating iTUG with other clinical scales with greater predictive value should also be conducted”.

Also, we added some changes in the discussion between lines 379-384, 389-416 and 469-488, as well abstract and conclusion were rewritten, between lines 1-11 and 515-523, respectively.  

Again, we appreciate the suggestions and hope that our modifications can improve the quality and understanding of our manuscript, in order to comply with the standard of the journal.

Sincerely,

Pablo Aqueveque Navarro, PhD.

Full Professor

Electrical Engineering Department,

Faculty of Engineering

Universidad de Concepción.

Concepción, Chile

Round 2

Reviewer 3 Report

I accept the authors' answer. The recommendations made have been taken into account. No additional comments.